# Ethnobotanical Survey of Local Flora Used for Medicinal Purposes among Indigenous People in Five Areas in Lagos State, Nigeria

**DOI:** 10.3390/plants11050633

**Published:** 2022-02-25

**Authors:** Ibraheem Oduola Lawal, Basirat Olabisi Rafiu, Joy Enitan Ale, Onuyi Emmanuel Majebi, Adeyemi Oladapo Aremu

**Affiliations:** 1Biomedicinal Research Centre, Forestry Research Institute of Nigeria, P.M.B. 5054, Jericho Hill, Ibadan 200272, Oyo, Nigeria; rafiu.bo@frin.gov.ng (B.O.R.); ale.je@frin.gov.ng (J.E.A.); 2Department of General Studies, Federal Cooperative College, P.M.B. 5033, Eleyele, Ibadan 200284, Oyo, Nigeria; majebioe@fccibadan.edu.ng; 3Indigenous Knowledge Systems Centre, Faculty of Natural and Agricultural Sciences, North-West University, Private Bag X2046, Mmabatho 2790, North West, South Africa

**Keywords:** ethnobotanical indices, Fabaceae, herbs, insomnia, malaria, respiratory-related diseases

## Abstract

Traditional medicine is typically the most accessible primary healthcare for a large proportion of the people in Nigeria. However, its potential remains under-explored, especially with regards to their documentation. This research investigated and documented the use of medicinal plants in the management of various health conditions/diseases among local populations in Lagos State. This study was conducted in five (5) locations of Lagos State i.e., Alimosho, Badagry, Eti-Osa, and Epe (including Ijebu and Imota). Ethnobotanical information from 100 participants was obtained using semi-structured questionnaires. Frequency of citation (FC), relative frequency of citation (RFC), fidelity level (FL), and informant consensus factor (ICF) were used to assess the importance of plants utilised for various health conditions/diseases. We identified 183 plants from 61 plant families with the highest number (24) of plants belonging to Fabaceae. Based on the high FC, the top-five popular plants used for managing health conditions/diseases in the study areas were *Mangifera indica* (95%), *Waltheria indica* (93%), *Zingiber officinale* (87%), *Alchornea cordifolia* (83%) and *Ipomoea involucrata* (81%). Furthermore, *Rauvolfia vomitoria*, *Urena lobata* and *Waltheria indica* were recognised as the most adaptable plants, as they were used to treat five different health conditions/diseases. The most commonly used life-forms were herbs (34%) and woody species (shrubs; 30%, and trees; 22%). The most regularly used plant parts were leaves. The calculated RFC values for all medicinal plant species ranged from 0.01 to 0.95, while FL values ranged from 7.14 to 100%. We found 14 health conditions/diseases, with ICF values ranging from 0.88 to 0.95. Insomnia, insanity, convulsion, nervousness, and muscle relaxants had the lowest (ICF = 0.88) agreement, while malaria/fevers, stomach, and respiratory-related diseases had the most (ICF = 0.95) agreement. The documented therapeutic uses of the plants provide basic data for further research aimed at pharmacological and conservation studies of the most important flora existing in the study areas.

## 1. Introduction

Globally, intense efforts are being geared towards the documentation of plant resources, with the goal of ensuring their sustainable utilisation and conservation to meet the needs of humans [1,2,3,4,5,6]. Particularly, the need to explore, document and preserve the indigenous knowledge associated with medicinal plants has been identified as one of the seven priorities for strategic action (Shenzhen’s proclamation of 2017) among plant scientists [7]. Furthermore, indigenous knowledge embedded in traditional medicine, including the use of medicinal plants, often offers culturally familiar techniques that address both the physical and spiritual state of an individual [8,9]. The inherent benefits such as relative affordability and accessibility of traditional medicine to a significant portion of the global population, especially in developing countries, further highlight the renewed interest from different stakeholders [4,5,10]. An increasing number of ethnobotanical studies are being done among diverse tribes and ethnic groups around the world, based on the understanding and acknowledgement of the relevance of plants for therapeutic effects and associated cultural value [11,12,13,14,15,16,17,18,19,20,21].

In Nigeria, the diverse floristic compositions of plant forms, including trees, shrubs, herbs, and other non-timber forest resources, has contributed greatly to the widespread use of Nigerian plants as medicine [1,13,16,22,23,24]. Furthermore, the use of medicinal plants is well-known among the indigenous peoples of Southwestern Nigeria. Particularly in Lagos State, ethnobotanical studies have been approached from different perspectives including the focus on plant species used for medicinal purposes by indigenous residents known as the Eegun people [25] as well as generating plant inventory used against various health conditions such as diabetics, hypertension, erectile dysfunction and fever [26,27].

Despite the increasing interest and commitment toward the documentation of plant resources with therapeutic value, some research gaps still exist in Nigeria including Lagos State [1,22]. Substantial documentation remains important to mitigate the potential loss of valuable indigenous knowledge associated with plant resources among local communities. In addition, the socio-cultural uniqueness and inherent dynamics associated with understanding local and customary medicinal use among local communities cannot be underestimated [9,23,28]. The potential of medicinal plants is still far from being extensively explored locally and nationally as well as internationally. Thus, the aim of this study was to generate an inventory of plants with medicinal value among selected local communities in Lagos State.

## 2. Results and Discussion

### 2.1. Inventory of Plant Species Used to Treat a Variety of Health Conditions/Diseases

The current study identified 183 medicinal plants from 61 plant families used in treating a variety of human health problems and diseases (Appendix A; Figure 1). The distribution of the plants across the selected study sites revealed an increasing number of medicinal plants in the order of Eti-Osa (78), Imota (89), Ijebu (91), Badagry (93) and Alimosho (100) (Appendix A). In addition, these five aforementioned locations had 24 medicinal plants in common. In comparison to similar ethnobotanical studies conducted in Southwestern Nigeria [13,16,23,25,29], the current study areas had a higher number of plants used for medicinal purposes. The generated inventory contributes to a global effort to document local flora and their accompanying indigenous knowledge for the benefit of the current and future generations [1,7,22]. Furthermore, the current study adds to the existing database of valuable medicinal plants in Southwestern Nigeria [1].

Multiple vernacular/local names were reported for a significant number of the identified plants in the current study, indicating their relevance among the participants. Plants with more than three local names included *Malvastrum coromandelianum, Microdesmis puberula, Mariscus alternifolius, Senna tora* and *Urena lobata*. Mukaila et al. [16] highlighted the use of more than one local name for medicinal plants in a recent ethnobotanical survey in Southwestern Nigeria, and evidence of newly recorded vernacular names was reported. Plant naming is frequent among ethnic groups, and the same plant may be given multiple names in different languages and localities [1,17,30,31,32]. Local names for plants are not chosen at random, they typically reflect their socio-cultural and medicinal significance [33], and can serve as a point of reference for community members.

### 2.2. Plant Families Used for Medicinal Purpose

Fabaceae was the most dominant (approximately 13%) plant family utilised as medicine for various health conditions/diseases in the study area (Figure 1). Other well-represented plant families were Asteraceae (8%), Malvaceae (7.7%), Euphorbiaceae (7%) and Lamiaceae (5%). On the other hand, about 70% of the 61 plant families documented had 1–2 representative members used for medicinal purposes (Appendix A). The dominance of Fabaceae as the most abundant family has been widely observed in several ethnobotanical studies in Nigeria [13,24,29,34] and other African countries such as Ethiopia [19], Ghana [31] and South Africa [12]. Existing data support the popularity of Fabaceae for their therapeutic efficacy, according to an exhaustive assessment of plant groups utilised in African traditional medicine [35]. In addition, families such as Apocynaceae, Burseraceae and Rubiaceae are considered as some of the most commonly traded species of African medicinal plants, an indication of their commercial value [35]. The popularity and high preference of plants from Fabaceae in African traditional medicine can be attributed to their availability and abundance, as well as their adaptability to diverse environments.

### 2.3. Life-Forms and Utilised Parts of the Documented Medicinal Plants

The inventory of 183 medicinal plants was represented by diverse life-forms which were mainly dominated (about 51%) by woody species consisting of trees and shrubs (Figure 2). Woody plants are essential to the African landscape in terms of biodiversity [36,37]. In particular, a significant portion of woody plants is well represented in traditional African medicine. There is increasing evidence that they are used for medicinal purposes in local communities [38,39,40,41,42,43]. When compared with herbaceous species, the predominance of woody species in traditional medicine is related to their relatively longer availability and persistence of the different botanical components used [37,44].

In the study area, a significant portion (34.4%) of the medicinal plants were herbs (Figure 2). A similar high dependence on herbaceous plants was evident in some previous studies [16,19]. Globally, herbaceous life forms are often abundant and widely distributed thereby contributing to their frequent utilisation as herbal medicine among different communities. In some instances, the ease of harvesting and collecting herbs explains their dominance in traditional medicine.

Overall, leaves were the most frequently used plant part identified in this study (Figure 3). The dominance of leaves relative to the stem and roots were clearly evident in the majority of the five selected sites. However, preference of the stem over the leaves was slightly higher if not similar in Ijebu and Eti-Osa. From previous studies, the popularity of the leaves over other plant parts remains a common trend among local communities with a rich history of traditional medicine [16,17,31]. The high number of metabolic activities (including photosynthesis) that occurs in the leaves may have resulted in the build-up of valuable compounds with medicinal properties [31]. When compared to other plant parts, leaves are frequently readily available, making them easier to access and harvest. From the standpoint of conservation and sustainability, leaves are chosen over plant elements such as bark and roots [45]. In general, harvesting leaves exerts less strain on the regeneration of a plant than the use of roots and bark. The prominence of leaves in traditional medicine may be attributed to these aforementioned characteristics.

### 2.4. Ethnobotanical Indices of Plant Species Used to Treat a Variety of Health Conditions/Diseases

Ethnobotanical indices are frequently used to infer the value of plants found in a specific study area [46,47]. These indices are widely used to rank medicinal plants based on their value, cultural significance, and perceived efficacy, and they are also useful tools for setting conservation priorities and prospecting for potentially beneficial therapeutic chemicals [28,46]. However, the interpretation and utility of these indices are frequently disputed, particularly when sample sizes vary [16,21,28,46]. As a result, the findings must be weighed against the purpose and scope of the research. The relative value of the described medicinal plants was analysed in the current study using various ethnobotanical indices (Table 1). Some of these ethnobotanical indices (for example, FC and RFC) are generated from one another, resulting in a similar pattern [46].

Cough, malaria, impotence, and haemorrhage were the most commonly treated conditions in the study area. We used ICF to determine the importance of selected plants for medicinal purposes (Table 2). Several plants have been used to treat a variety of health problems/diseases. Based on their indications for the treatment of five health conditions/diseases, *Rauvolfia vomitoria*, *Urena lobata* and *Waltheria indica* have been identified as the most versatile plants.

#### 2.4.1. Frequency of Citation (FC, %) and Relative Frequency of Citation (RFC) for the Documented Plants

We recorded FC and RFC values that ranged from 1–95 and 0.01–0.95, respectively. An estimated 19.13% of the plants had a value of at least 50% (0.5). Examples of these plants included *Mangifera indica*, *Waltheria indica*, *Zingiber officinale*, *Alchornea cordifolia*, *Ipomoea involucrata*, *Ageratum conyzoides*, *Vernonia amygdalina*, *Ocimum gratissimum* and *Albizia zygia*. Given the significant number of mentions by the participants, these plants are well-known and play an important role in the maintenance of the health and well-being among community members. Plant species with higher FC and RFC values are regarded as more popular and well known among local inhabitants [20]. On the other hand, *Cucumeropsis mannii*, *Desmodium velutinum*, and *Euphorbia heterophylla* were the three least-mentioned plants by the participants.

#### 2.4.2. Fidelity Level (%) for the Documented Plants

As a reasonable proxy, FL may be relevant for identifying the preferred plant(s) for treating a certain ailment [48]. We examined each illness category with the highest degree of agreementto highlight the most essential plants utilised in each category. As shown in Table 1, the highest (100%) FL was recorded for *Artocarpus communis* (malaria), *Anthocleista djalonensis* (hypertension), *Anthocleista vogelii* (divinity), *Anacardium occidentale* (malaria), *Amaranthus viridis* (longevity), *Hippocratea pallens* (malaria), *Hoslundia opposita* (ulcer), *Sesamum indicum* (potency), *Sesamum radiatum* (potency), *Sida cordifolia* (cough), *Smilax kraussiana* (erection), *Sorghum bicolor* (malaria), *Sterculia tragacantha* (diabetes), *Syzygium guineense* (diabetes), *Trichilia monadelpha* (insomnia) and *Urena lobata* var. *glauca* (potency). On the other hand, the lowest level of FL was indicated for plants such as *Alchornea cordifolia* (malaria, 7.23%), *Lantana camara* (nervousness, 7.14%), *Borreria verticillata* (contraceptive, 11.63%) and *Emilia coccinea* (contraceptive, 8.11%).

#### 2.4.3. Informant Consensus Factor (ICF) in the Study Area

As a measure of the agreement among participants that a plant or group of plants species can cure a particular disease category, the importance of ICF cannot be overemphasised [49]. This may be particularly useful in selecting plants for pharmacological and phytochemical studies [18,50]. Even though it is often difficult to classify diseases from ethnographic surveys [9], it is nevertheless essential to understand local attitudes about disease and the use of medicinal plants. In traditional medicine, the use of plants has been well-described for health conditions such as metabolic-related diseases [51,52], malaria [53], respiratory conditions [54], pains and inflammations [55,56]. In this study, the health conditions were grouped into 14 categories and ICF values ranged from 0.88–0.95 (Table 2). In the study area, respiratory-related ailments, stomach-related ailments, malaria and other fevers, had the highest agreement among the participants with an ICF of 0.95. This was followed by infectious diseases, cardiovascular-related diseases, foot pain, neck pain, rheumatism, arthritis (ICF = 0.94). The least agreement among the participants was observed in insomnia, insanity, convulsion, nervousness, muscle relaxant category with an ICF value of 0.88.

The higher ICF scores for respiratory-related diseases, stomach-related disorders, malaria, and other fever categories show that there is a high level of agreement in the research area regarding the management of these health problems. The majority of the aforementioned health issues include a variety of symptoms that traditional healers may easily identify. This is most likely responsible for the enormous number of plants and, as a result, the higher ICF. Pharmacologically effective antidotes have been recognised to have higher ICF values [49]. Therefore, the results of ICF may be useful in prioritising plants for further pharmacological studies, as the efficacy of traditional herbal medicines is highly correlated with the value of ICF [18,20,50].

## 3. Materials and Methods

### 3.1. Study Area

This study was conducted in five (5) purposively selected locations of Lagos State, Nigeria (Alimosho, Badagry, Eti-Osa and Epe, which included Ijebu and Imota) (Figure 4). These sites are located at longitudes 20°42′ E and 32°2′ E, and latitudes 60°22′ N and 60°2′ N, respectively. Lagos State is surrounded by Ogun State to the north and east and by the Republic of Benin to the west. It stretches for more than 180 km along the Guinean coast in the Bight of Benin of the Atlantic Ocean. Its political territory and jurisdiction include the city of Lagos and the four administrative divisions of Badagry, Epe, Ikeja and Ikorodu. It covers an area of 358,862 hectares (3577 km^2^), representing 0.4% of Nigeria’s total land area of 923,773 m^2^ [57]. The inhabitants were Yoruba, but the first settlers were Awori hunters and fishermen who migrated from Ile-Ife to the coast as well as the Egbas, who are known for their carvings, sculptures, and traditional arts [58]. The main vegetation in Lagos State is freshwater swamp forest and mangrove swamp forest, both of which are affected by the double rainfall pattern, which makes up the wetland ecosystem. In Lagos State, there are two distinct seasons: dry (November–March) and wet (April–October) [59].

### 3.2. Field Interview

Following project approval by the study ethics committee, participants were briefed on the purpose of the survey in order to obtain their consent and willingness to participate. We recruited ten (10) native field assistants to help with administration and explaining the questions to participants (in Yoruba, the local language). The ethnobotanical survey was carried out between July 2018 and December 2019.

Using a semi-structured questionnaire, we collected ethnobotany information from 100 randomly selected participants (herbal vendors, traditionalists, farmers, hunters) (Appendix A). The majority of participants (89%) were male and basic education (79%) was predominantly their highest level of education (Table 3). Participants were asked about their understanding of the use of plant species in the treatment of various health conditions/diseases in the region. In addition, we documented information such as plant parts, administration, preparation, and dosages used for the stated health problem/diseases.

### 3.3. Plant Collection and Identification

Plant specimens from the research area were collected based on their vernacular names. An expert recognised and authenticated voucher specimens for all of the plants, which were prepared according to taxonomic norms. These plants were then donated to the Forest Herbarium Ibadan (FHI), Forestry Research Institute of Nigeria. Given the importance of precise scientific names [30,32], the current taxonomic classification was validated using the website ‘The Plant List’ (http://www.theplantlist.org/, accessed on 19 February 2022).

### 3.4. Quantitative Ethnobotanical Indices

To establish the importance of the identified plants against various diseases among the participants, we analysed the data using the following ethnobotanical indices.

#### 3.4.1. Frequency of Citation (FC) and Relative Frequency of Citation (RFC)

We calculated the frequency of citation (FC) and relative frequency of citation (RFC) for the plant species mentioned in the search areas using the method published by Tardío and Pardo-de-Santayana [60]. Accordingly, FC and RFC were calculated as follows:FC = (number of times a particular species is mentioned)/(total number of times all species are mentioned) × 100.
RFC = FC/N.
where FC is the number of informants referring to species use and N denotes the number of informants who participated in the survey.

This index, in theory, ranges from 0 to 1. If the RFC index is zero, no one cites valuable plants; if the RFC index is one, all informants in the study mention useful plants.

#### 3.4.2. Fidelity Level (FL, %)

The fidelity level (FL) is used to indicate the percentage of participants who stated the use of certain plant species for the same medicinal purpose [48]. FL was calculated as follows:FL (%) = (Np/N) × 100
where Np: the number of participants who confirm the use of a plant species to treat certain diseases. N: the number of participants who use plants as medicine to cure several diseases.

#### 3.4.3. Informant Consensus Factor (ICF)

For the general analysis of plant use data, the informant consensus factor (ICF) was used to verify the homogeneity of the information obtained [61]. All citations are placed in the category of disease for which the plant is claimed to be used.

ICF = (Nur − Nt)/(Nur − 1)
where Nur: number of usage citations in each category and Nt: number of species used. The ICF value will be low (close to 0) if the plant is selected at random or if the whistle-blower does not exchange information about its use and the value will be high (close to 1) if there are well-defined selection criteria in the community and/or if the information is shared among informants.

### 3.5. Ethical Consideration

The study was approved (CFGO711FRIN06) by the Ethics Committee of Forestry Research Institute of Nigeria, Ibadan, Nigeria. Access to the study areas was provided by the traditional authority. All the participants provided consent prior to the study.

## 4. Conclusions

The current research revealed 183 medicinal plants with diverse therapeutic uses in the study areas. This generated inventory contributes to local, national and international efforts aimed at documenting indigenous flora with medicinal benefits among local communities. Given their high mentions (FC, RFC), *Mangifera indica*, *Waltheria indica* and *Zingiber officinale* were identified as the most popular plants used for medicinal purposes among the participants. The most versatile plants, given their utilisation for managing at least five health conditions/diseases, included *Rauvolfia vomitoria*, *Urena lobata* and *Waltheria indica*. In relation to the ICF, we generated 14 health conditions/diseases managed with the use of medicinal plants in the study areas. As an indication of the vital role of traditional medicine for healthcare needs, the highest degree of consensus on plants was evident for malaria, stomach, and respiratory-related conditions. These aforementioned health conditions/diseases are known to be prevalent in rural areas and the use of medicinal plants for managing them highlights the importance of traditional medicine for primary healthcare needs. Taken together, the documented therapeutic uses of these plants provide baseline data for further research aimed at pharmacological and conservation studies.

## Figures and Tables

**Figure 1 plants-11-00633-f001:**
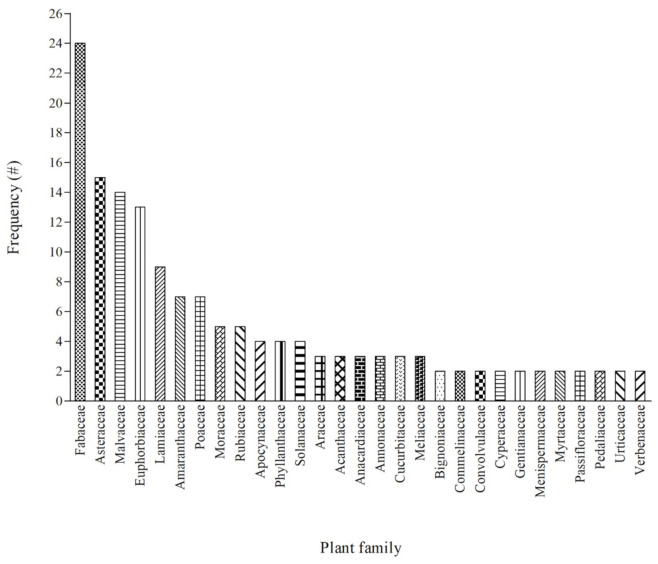
Frequency of plant families used for the management of diverse health conditions/diseases in five (5) selected locations in Lagos State of Nigeria.

**Figure 2 plants-11-00633-f002:**
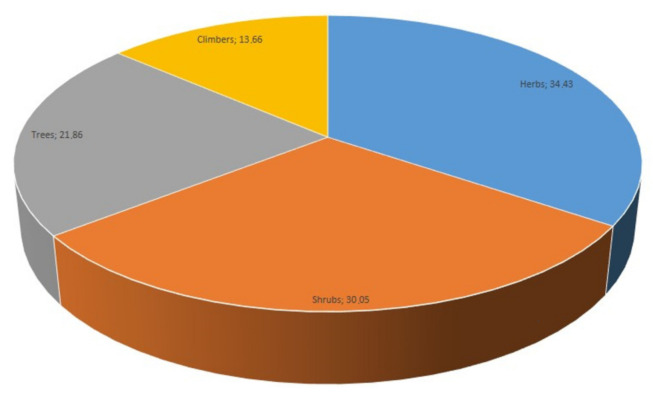
Life-form distribution (%) for the plant species used for managing health conditions/diseases in five (5) selected locations in Lagos State of Nigeria.

**Figure 3 plants-11-00633-f003:**
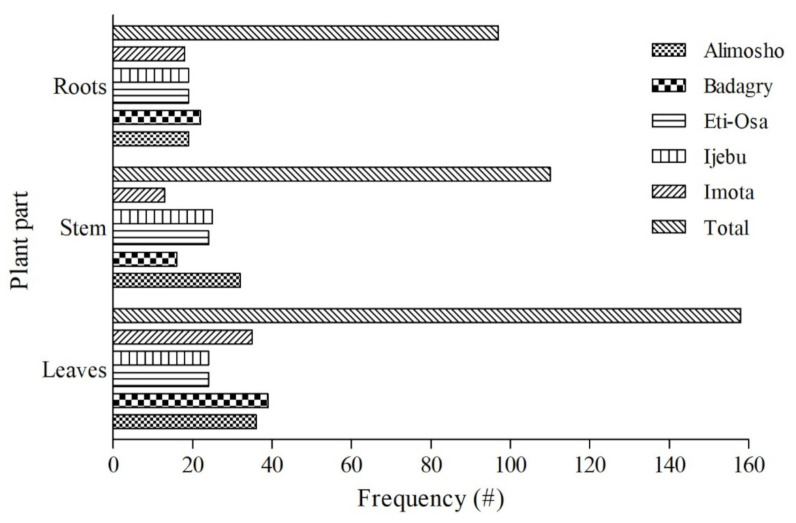
Frequency of plant parts used for the management of diverse health conditions/diseases in five (5) selected locations in Lagos State of Nigeria.

**Figure 4 plants-11-00633-f004:**
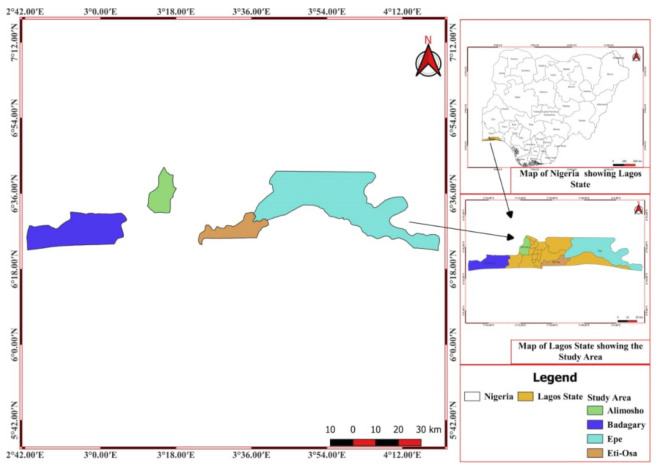
Locations selected for the ethnobotanical survey conducted in Lagos State of Nigeria.

**Table 1 plants-11-00633-t001:** Fidelity level (FL, %), relative frequency of citation (RFC) and frequency of citation (FC) for plants used to treat a variety of health conditions/diseases in five (5) selected locations in Lagos State of Nigeria.

Plant Species	Heath Condition, FL (%)	RFC	FC (%)
*Abrus precatorius* L.	Cough, 71.70 Stomachache, 28.30	0.53	53
*Acalypha fimbriata* Schumach. & Thonn.	Cough, 38.7 Divinity, 61.29	0.31	31
*Acanthospermum hispidum* DC.	Hypertension, 62.96 Divinity, 37.04	0.27	27
*Achyranthes aspera* L.	Erection, 28.17 Sight, 47.89 Potency, 23.94	0.71	71
*Adenia lobata* (Jacq.) Engl.	Piles, 33.33 Arthritis, 66.67	0.48	48
*Ageratum conyzoides* (L.) L.	Infectious disease, 21.25 Malaria, 42.5 Hypertension, 20.00 Sight, 16.25	0.80	80
*Albizia ferruginea* (Guill. & Perr.) Benth.	Cough, 46.15 Pregnancy, 53.85	0.26	26
*Albizia lebbeck* (L.) Benth.	Fever, 21.05 Arthritis, 45.61 Cough, 33.33	0.57	57
*Albizia zygia* (DC.) J.F.Macbr.	Aphrodisiac, 25.97 Arthritis, 53.25 Cough, 33.77 Toothache, 3.90	0.77	77
*Alchornea cordifolia* (Schumach. & Thonn.) Müll.Arg.	Arthritis, 18.07 Sight, 39.76 Ulcers, 15.66 Malaria, 7.23	0.83	83
*Alchornea laxiflora* (Benth.) Pax & K.Hoffm.	De-worming, 24.56 Infectious diseases, 10.53 Malaria, 35.09 Oral hygiene, 29.82	0.57	57
*Alternanthera sessilis* (L.) R.Br. ex DC.	Sight, 64.71 Jaundice, 35.29	0.17	17
*Amaranthus spinosus* L.	Sight, 42.86 Cirrhosis, 57.14	0.21	21
*Amaranthus viridis* L.	Longevity, 100.00	0.13	13
*Anacardium occidentale* L.	Malaria, 100.00	0.38	38
*Anchomanes difformis* (Blume) Engl.	Chicken pox, 40.00 Measles, 60.00	0.20	20
*Annona senegalensis* Pers.	Malaria, 53.33 Potency, 46.67	0.30	30
*Anthocleista djalonensis* A.Chev.	Hypertension, 100.00	0.26	26
*Anthocleista vogelii* Planch.	Divinity, 100.00	0.07	7
*Artocarpus communis* J.R.Forst. & G.Forst. (Synonym: *Artocarpus altilis* (Parkinson ex F.A.Zorn) Fosberg)	Malaria, 100.00	0.20	20
*Aspilia africana* (Pers.) C.D.Adams	Fibroid, 26.00 Cough, 56.00 Purgative, 18.00	0.50	50
*Asystasia gangetica* (L.) T.Anderson	Neck pain, 12.50 Sight, 43.75 Potency, 15.63 Cough, 28.13	0.64	64
*Azadirachta indica* A.Juss.	Anti-snake bite, 100.00	0.16	16
*Baphia nitida* Lodd.	Divinity, 100.00	0.12	12
*Barleria opaca* (Vahl) Nees	Diabetes, 100.00	0.17	17
*Bidens pilosa* L.	Cough, 100.00	0.19	19
*Boerhavia diffusa* L.	Malaria, 100.00	0.18	18
*Borreria scabra* (Schumach. & Thonn.) K.Schum. (Synonym: *Spermacoce ruelliae* DC.)	Ringworm, 19.18 Diabetes, 35.62 Eczema, 21.92 Birth control, 23.28	0.73	73
*Borreria verticillata* (synonym: *Spermacoce verticillata* L.)	Ringworm, 88.37 Birth control, 11.63	0.43	43
*Bridelia ferruginea* Benth.	Diabetes, 66.04 Birth control, 33.96	0.53	53
*Bryophyllum pinnatum* (Lam.) Oken	Cough,72.22 Sight, 27.78	0.54	54
*Caladium bicolor* (Aiton) Vent.	Pimples, 61.70 Cough, 38.30	0.47	47
*Calophyllum inophyllum* L.	Scurvy, 100	0.08	8
*Calopogonium mucunoides* Desv.	Cough, 60.94 Scurvy, 17.19 Ulcer, 21.88	0.64	64
*Calotropis procera* (Aiton) Dryand.	Malaria, 60.00 Conjunctivitis, 40.00	0.25	25
*Canna indica* L.	Cirrhosis, 50.00 Pimples, 50.00	0.28	28
*Canavalia ensiformis* (L.) DC.	Pregnancy, 100.00	0.10	10
*Carpolobia lutea* G.Don	Easy labour, 100.00	0.12	12
*Celosia argentea* L.	Malaria, 100.00	0.02	2
*Centrosema pubescens* Benth.	Pimples, 46.15 Purgative, 53.85	0.13	13
*Chassalia kolly* (Schumach.) Hepper	Aphrodisiac, 100.00	0.18	18
*Chromolaena odorata* (L.) R.M.King & H.Rob.	Malaria, 93.55 Potency, 6.45	0.62	62
*Cissampelos owariensis* P.Beauv. ex DC.	Miscarriage, 100.00	0.36	36
*Citrus aurantiifolia* (Christm.) Swingle	Malaria, 100.00	0.26	26
*Cleistopholis patens* (Benth.) Engl. & Diels	Pimples, 100.00	0.23	23
*Cleome fruticosa* L. (Synonym: *Cadaba fruticosa* (L.) Druce)	Cirrhosis, 61.90 Birth control, 38.10	0.21	21
*Clerodendrum capitatum* (Willd.) Schumach. & Thonn	Malaria, 53.85 Diabetes, 46.15	0.26	26
*Clerodendrum paniculatum* L.	Cirrhosis, 100.00	0.06	6
*Clerodendrum umbellatum* Poir.	Stomachache, 100.00	0.16	16
*Clerodendrum volubile* P.Beauv.	Immune booster, 100.00	0.02	2
*Cnestis ferruginea* Vahl ex DC.	Immune booster, 100.00	0.05	5
*Cola millenii* K.Schum.	Miscarriage, 100.00	0.09	9
*Colocasia esculenta* (L.) Schott	Cough, 100.00	0.14	14
*Commelina africana* L.	Diabetes, 78.57 Curse, 21.43	0.14	14
*Commelina erecta* L.	Potency, 100.00	0.15	15
*Costus afer* Ker Gawl.	Cough, 58.44 Ulcer, 41.56	0.77	77
*Croton lobatus* L. (Synonym: *Astraea lobata* (L.) Klotzsch).	Cough, 100.00	0.18	18
*Croton zambesicus* Müll.Arg. (Synonym: *Croton gratissimus* Burch.)	Malaria, 56.52 Hypertension, 43.48	0.23	23
*Cucumeropsis mannii* Naudin	Cough, 100.00	0.01	1
*Cyathula prostrata* (L.) Blume	Pimples, 100.00	0.03	3
*Cymbopogon citratus* (DC.) Stapf	Longevity, 100.00	0.09	9
*Cyperus haspans* L.	Arthritis, 100.00	0.11	11
*Dalbergia saxatilis* Hook.f.	Longevity, 100.00	0.14	14
*Datura metel* L.	Hypertension, 48.27 Immune booster, 37.93 Muscle relaxant, 13.80	0.29	29
*Desmodium velutinum* (Willd.) DC.	Erection, 100.00	0.01	1
*Dichrostachys cinerea* (L.) Wight & Arn.	Headache, 27.03 Toothache, 32.43 Cough, 40.54	0.37	37
*Dissotis rotundifolia* (Sm.) Triana (Synonym *Heterotis rotundifolia* (Sm.) Jacq.-Fél.)	Easy labour, 20.69 Pregnancy care, 79.31	0.29	29
*Eclipta prostrata* (L.) L.	Cough, 100.00	0.16	16
*Elaeis guineensis* Jacq.	Blood tonic, 100.00	0.03	3
*Eleusine indica* (L.) Gaertn.	Pile, 19.57 Cough, 43.47 Malaria, 36.96	0.46	46
*Eleutheranthera ruderalis* (Swartz) Sch.-Bip.	Piles, 61.54 Longevity, 38.46	0.26	26
*Emilia coccinea* (Sims) G.Don	Foot pain, 59.46 Birth control, 8.11 Purgative, 32.43	0.37	37
*Entandrophragma angolense* (Welw.) C.DC.	Piles, 100.00	0.15	15
*Eragrostis namaquennsis* Nees ex Schrad. (Synonym: *Eragrostis japonica* (Thunb.) Trin.)	Stomachache, 100.00	0.10	10
*Erigeron floribundus* (Kunth) Sch.Bip.	Scurvy, 100.00	0.09	9
*Erythrina senegalensis* DC.	Diabetes, 100.00	0.13	13
*Euphorbia glaucophylla* Poir. (Synonym: *Euphorbia trinervia* Schumach. & Thonn.)	Longevity, 46.94 Malaria, 22.45 Cough, 30.61	0.49	49
*Euphorbia heterophylla* L.	Longevity, 100.00	0.01	1
*Ficus benjamina* L.	Scurvy, 30.30 Cough, 69.70	0.33	33
*Ficus capensis* Thunb. (Synonym: *Ficus sur* Forssk.)	Hypertension, 26.67 Cough, 73.33	0.45	45
*Ficus exasperata* Vahl	Scurvy, 50.00 Cough, 27.27 Hypertension, 22.73	0.44	44
*Ficus polita* Vahl	Pimples, 100.00	0.06	6
*Fleurya aestuans* (L.) Chew	Cough, 100.00	0.18	18
*Gliricidia sepium* (Jacq.) Walp.	Arthritis, 46.15 Hypertension, 53.85	0.26	26
*Glyphaea brevis* (Spreng.) Monach.	Ulcer, 100.00	0.13	13
*Gomphrena celosioides* Mart.	Pimples, 62.16 Potency, 37.84	0.37	37
*Grewia pubescens* P.Beauv.	Hypertension, 100.00	0.16	16
*Harungana madagascariensis* Lam. ex Poir.	Anti-snake bite, 100.00	0.18	18
*Heliotropium indicum* L.	Erection, 33.33 Piles, 66.67	0.27	27
*Hibiscus rosa-sinensis* L.	Cough, 100.00	0.10	10
*Hibiscus surattensis* L.	Diabetes, 65.71 Potency, 34.29	0.35	35
*Hippocratea pallens* Planch. ex Oliv. (Synonym: *Apodostigma pallens* (Planch. ex Oliv.) R.Wilczek)	Malaria, 100.00	0.17	17
*Hoslundia opposita* Vahl	Ulcer, 100.00	0.19	19
*Hyptis suaveolens* (L.) Poit.	Anti-snake bite, 100.00	0.22	22
*Icacina trichantha* Oliv.	Curse, 20.90 Potency, 79.10	0.67	67
*Indigofera arrecta* A.Rich.	Piles, 61.11 Pregnancy, 38.89	0.18	18
*Indigofera hirsuta* L.	Sight, 22.95 Stomachache, 45.90 Purgative, 31.15	0.61	61
*Ipomoea involucrata* P.Beauv. (Synonym: *Ipomoea pileata* Roxb.)	Anti-snake bite, 19.75 Malaria, 19.76 Cough, 60.49	0.81	81
*Jatropha curcas* L.	Piles, 83.64 Sight, 16.36	0.55	55
*Jatropha gossypiifolia* L.	Piles, 100.00	0.40	40
*Kigelia africana* (Lam.) Benth.	Cough, 46.43 Piles, 53.57	0.28	28
*Lantana camara* L.	Purgative, 50.00 Nervousness, 7.14 Potency, 42.86	0.14	14
*Lawsonia inermis* L.	Malaria, 77.27 Gonorrhoea, 22.73	0.44	44
*Leucaena leucocephala* (Lam.) de Wit	Purgative, 100.00	0.06	6
*Luffa cylindrica* (L.) M.Roem.	Cramps, 14.00 Fever, 30.00 Purgative, 38.00 Convulsion, 18.00	0.50	50
*Macaranga barteri* Müll.Arg.	Easy labour, 42.42 Anti-snake bite, 57.58	0.33	33
*Microdesmis puberula* Hook.f. ex Planch.	Easy labour, 100.00	0.11	11
*Mallotus oppositifolius* (Geiseler) Müll.Arg.	Stomach problems, 51.61 Malaria, 32.26 Sight, 16.13	0.31	31
*Malvastrum coromandelianum* (L.) Garcke	Anti-snake bite, 100.00	0.02	2
*Mangifera indica* L.	Malaria, 76.84 Fever, 23.16	0.95	95
*Margaritaria discoidea* (Baill.) G.L.Webster	Malaria, 100.00	0.23	23
*Mariscus alternifolius* Vahl (Synonym: *Cyperus cyperoides* (L.) Kuntze)	Jaundice, 33.33 Arthritis, 66.67	0.09	9
*Melanthera scandens* (Schumach. & Thonn.) Roberty	Curse, 37.50 Malaria, 62.50	0.24	24
*Merremia pterygocaulos* (Choisy) Hallier f.	Foot pain, 80.00 Sight, 20.00	0.20	20
*Mezoneuron benthamianum* (Synonym: *Caesalpinia benthamiana* (Baill.) Herend. & Zarucchi)	Piles, 100.00	0.25	25
*Mimosa pudica* L.	Erection, 84.00 Anti-snake bite, 16.00	0.25	25
*Morinda lucida* Benth.	Malaria, 62.86 Fever, 24.29 Jaundice, 12.85	0.70	70
*Moringa oleifera* Lam.	Cough, 100.00	0.27	27
*Myrianthus arboreus* P.Beauv.	Neck pain, 70.59 Cough, 29.41	0.17	17
*Nauclea latifolia* Sm. (Synonym: *Sarcocephalus latifolius* (Sm.) E.A.Bruce)	Malaria, 50.00 Pimples, 27.27 Diabetes, 22.73	0.66	66
*Newbouldia laevis* (P.Beauv.) Seem.	Diabetes, 40.30 Measles, 25.37 Worm-expellant, 10.45 Jaundice, 23.88	0.67	67
*Ocimum gratissimum* L.	Piles, 38.46 Purgative, 46.15 Cough, 15.39	0.78	78
*Panicum scandens* (Schrad. ex Schult.) Trin. (Synonym: *Setaria scandens* Schrad.)	Malaria, 100.00	0.21	21
*Passiflora foetida* L.	Foot pain, 80.65 Sight, 19.35	0.31	31
*Paullinia pinnata* L.	Aphrodisiac, 52.17 Piles, 47.83	0.23	23
*Perotis indica* (L.) Kuntze	Blood tonic,100.00	0.03	3
*Persea americana* Mill.	Longevity, 100.00	0.14	14
*Phaulopsis falcisepala* C.B.Clarke (Synonym: *Phaulopsis ciliata* (Willd.) Hepper)	Malaria, 100.00	0.15	15
*Phyllanthus amarus* Schumach. & Thonn.	Fever, 23.40 Pregnancy, 36.18 Immune booster, 17.02 Cough, 23.40	0.47	47
*Phyllanthus niruri* L.	Foot pain, 57.89 Sight, 42.11	0.19	19
*Physalis angulata* L.	Skin rashes, 76.00 Piles, 24.00	0.50	50
*Piliostigma thonningii* (Schum.) Milne-Redh. (Synonym: *Bauhinia thonningii* Schum.)	Potency, 100.00	0.06	6
*Pinus caribaea* Morelet	Cough, 100.00	0.05	5
*Pleioceras barteri* Baill.	Neck pain, 38.89 Stomachache, 61.11	0.18	18
*Polyalthia suaveolens* Engl. & Diels (Synonym: *Greenwayodendron suaveolens* (Engl. & Diels) Verdc.)	Cough, 100.00	0.02	2
*Portulaca oleracea* L.	Cough, 55.56 Anti-snake bite, 44.44	0.18	18
*Psidium guajava* L.	Fever, 22.50 Stomachache, 77.50	0.40	40
*Rauvolfia vomitoria* Afzel.	Malaria, 16.13 Insanity, 29.03 Hypertension, 22.58 Muscle relaxant, 16.13 Cough, 16.13	0.62	62
*Ricinus communis* L.	Divinity, 44.83 Sight, 55.17	0.29	29
*Scoparia dulcis* L.	Black coated tongue, 30.77 Birth control, 69.23	0.13	13
*Secamone afzelii* (Roem. & Schult.) K.Schum.	Immune booster, 100.00	0.26	26
*Securinega virosa* (Roxb. ex Willd.) Baill. (Synonym: *Flueggea virosa* (Roxb. ex Willd.) Royle)	Typhoid, 47.83 Pregnancy, 28.26 Immune booster, 23.91	0.46	46
*Senna hirsuta* (L.) H.S.Irwin & Barneby	Cough, 37.04 Purgative, 62.96	0.27	27
*Senna obtusifolia* (L.) H.S.Irwin & Barneby	Scurvy, 39.02 Cough, 60.98	0.41	41
*Senna podocarpa* (Guill. & Perrottet) Lock	Stomach problem, 100.00	0.16	16
*Senna siamea* (Lamarck) H.S.Irwin & Barneby	Immune booster, 100.00	0.11	11
*Senna tora* (L.) Roxb.	Potency, 33.33 Sight, 19.44 Malaria, 36.12 Divinity, 11.11	0.36	36
*Sesamum indicum* L.	Potency, 100.00	0.03	3
*Sesamum radiatum* Schumach. & Thonn.	Potency, 100.00	0.06	6
*Sida acuta* Burm.f.	Fibroids, 38.89 Sight, 61.11	0.18	18
*Sida cordifolia* L.	Cough, 100.00	0.12	12
*Sida linifolia* Juss. ex Cav.	Scurvy, 58.82 Pimples, 41.18	0.17	17
*Smilax kraussiana* Meisn. (Synonym: *Smilax anceps* Willd.)	Erection, 100.00	0.12	12
*Solanum nigrum* L. (Synonym: *Solanum americanum* Mill.)	Diabetes, 46.87 Cough, 53.13	0.32	32
*Solanum torvum* Sw.	Pimples, 22.22 Potency, 41.67 Hypertension, 36.11	0.36	36
*Solenostemon monostachyus* (P.Beauv.) Briq. (Synonym: *Plectranthus monostachyus* (P.Beauv.) B.J.Pollard)	Piles, 26.92 Malaria, 34.62 Purgative, 17.31 Potency, 21.15	0.52	52
*Sorghum bicolor* (L.) Moench	Malaria, 100.00	0.07	7
*Sphenocentrum jollyanum* Pierre	Deworming, 45.16 Malaria, 19.36 Fever, 35.48	0.31	31
*Spigellia anthelmia* L.	Diabetes, 38.46 Piles, 61.54	0.26	26
*Spondias mombin* L.	Penile erection, 40.00 Diabetes, 28.57 Anti-snake bite, 14.29 Insomnia, 17.14	0.35	35
*Sporobolus indicus* (L.) R.Br.	Diabetes, 60.87 Purgative, 39.13	0.23	23
*Stachytarpheta indica* (L.) Vahl	Erection, 20.00 Hypertension, 80.00	0.20	20
*Sterculia tragacantha* Lindl.	Diabetes, 100.00	0.13	13
*Synedrella nodiflora* (L.) Gaertn.	Potency, 55.55 Sight, 5.56 Longevity, 27.78 Malaria, 11.11	0.18	18
*Syzygium guineense* (Willd.) DC.	Diabetes, 100.00	0.12	12
*Talinum triangulare* (Jacq.) Willd. (Synonym: *Talinum fruticosum* (L.) Juss.)	Blood tonic, 86.36 Malaria, 13.64	0.44	44
*Tapinanthus globiferous* (A.Rich.) Tiegh.	Fibroid, 100.00	0.11	11
*Telfairia occidentalis* Hook.f.	Blood tonic, 79.55 Hypertension, 20.45	0.44	44
*Terminalia ivorensis* A.Chev.	Divinity, 100.00	0.09	9
*Tetracera alnifolia* Willd.	Fibroid, 100.00	0.05	5
*Thaumatococcus daniellii* (Benn.) Benth.	Ulcer, 75.38 Food sweetener, 24.62	0.65	65
*Thevetia peruviana* (Pers.) K.Schum.	Purgative, 61.54 Pimples, 17.95 Cough, 20.51	0.39	39
*Tithonia diversifolia* (Hemsl.) A.Gray	Malaria, 86.15 Piles, 13.85	0.65	65
*Trichilia monadelpha* (Thonn.) J.J.de Wilde	Insomnia, 100.00	0.02	2
*Tridax procumbens* (L.) L.	Ulcer, 22.22 Foot pain, 33.33 Malaria, 19.44 Hypertension, 25.01	0.36	36
*Triumfetta cordifolia* A.Rich.	Easy labour, 64.29 Birth control, 35.71	0.14	14
*Urena lobata* L. *var.* var. *glauca* (Blume) Borss. Waalk.	Potency, 100.00	0.15	15
*Urena lobata* L.	Birth control, 18.64 Rheumatism, 16.95 Wound, 28.81 Diarrhoea, 22.03 Stomachache, 13.57	0.59	59
*Vernonia amygdalina* Delile	Diabetes, 18.99 Malaria, 39.24 Oral hygiene, 21.52 Diarrhoea, 20.25	0.79	79
*Vernonia cinerea* (L.) Less. (Synonym: *Cyanthillium cinereum* (L.) H.Rob.	Asthma, 28.57 Bronchitis, 22.45 Cold, 26.53 Stomachache, 22.45	0.49	49
*Vitex doniana* Sweet	Dysentery, 36.20 Diarrhoea, 24.14 Hypertension, 24.14 Indigestion, 15.52	0.58	58
*Waltheria indica* L.	Wound, 13.98 Blood tonic, 33.33 Ulcer, 17.21 Cold, 10.75 Cough, 24.73	0.93	93
*Zingiber officinale* Roscoe	Ulcer, 24.14 Indigestion, 27.59 Cough, 36.78 Flu, 11.49	0.87	87

**Table 2 plants-11-00633-t002:** Informant Consensus Factor (ICF) for medicinal plants from five (5) selected locations in Lagos State of Nigeria.

S/n	Disease Category	Number of Species	Use Citation	ICF
1	Respiratory-related ailments	46	842	0.95
2	Stomach-related ailments	50	917	0.95
3	Infectious diseases	21	320	0.94
4	Malaria and other fevers	49	944	0.95
5	Cardiovascular-related diseases	31	496	0.94
6	Male reproductive-related issues	27	304	0.91
7	Female reproductive-related issues	25	293	0.92
8	Vision-related issues	18	224	0.92
9	Foot pain, neck pain, rheumatism, arthritis	16	266	0.94
10	Oral hygiene, black coated tongue, toothache	5	53	0.92
11	Divinity, curse (spiritual-related)	10	100	0.91
12	Anti-snake bite, de-wormer, worm-expellant	10	118	0.92
13	Cirrhosis, scurvy, blood tonic, longevity, immune booster	31	404	0.93
14	Insomnia, insanity, convulsion, nervousness, muscle relaxant	7	50	0.88

**Table 3 plants-11-00633-t003:** Demographic characteristics of the participants (*n* = 100) in five (5) selected areas in Lagos State of Nigeria.

Feature	Frequency (*n*)
Age group	
30–39	2
40–49	28
50–59	34
60–69	18
70–79	12
80 and above	6
Gender	
Male	89
Female	11
Marital status	
Widower/widow	7
Single	2
Married	91
Religion status	
African traditionalist	60
Christianity	12
Islam	28
Formal education level	
None	7
Primary	79
Secondary	12
Tertiary	2
Occupation-type	
Farmer	29
Herbalist	52
Trader	19

## Data Availability

We have included all data related the study in the manuscript and as Appendix A.

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
