# Peer review of "Ethnobotanical Survey of Local Flora Used for Medicinal Purposes among Indigenous People in Five Areas in Lagos State, Nigeria"

_plants, 2022, doi:10.3390/plants11050633_

Round 1
Reviewer 1 Report
I recommended the publication of this manuscript after making of the revisions in the attached file

Reviewer 2 Report
The authors described the importance of medicinal plants in Nigeria. They have identified 183 species of medicinal plants from 61 plant families with Fabaceae having 24 higher number (24) of plants. The use most common described are herbs and woody species 25 (shrubs and trees).
The manuscript is a description of these plants in a great area of Nigeria. This fact corroborated the use and distributed the plant in all Nigeria local. This study revealed the diverse medicinal plants with therapeutic potential among indigenous people in five areas in Lagos State of Nigeria.
The manuscript is interesting in terms of dissemination and clarification of the most important plants for use by the population of Nigeria and should be accepted for publication in the Plants Journal.
Reviewer 3 Report
Unfortunately, I am forced recommending to reject the manuscript. The main reasons are listed below.
However, the title, abstract, and keywords are written clearly. Just the key word "conservation" is recommended to be deleted because no conservation things are present in the paper.
In general, the manuscript has a descriptive character. There is no analysis of the obtained, just their listing in the form of tables and graphs. The largest part of the section Results and Discussion looks like Results with certain fragments of discussion and comparisons with the known literature. The sampling amount of n=100 is certainly low to obtain reliable results, taking into account that the study has covered 1.5 years (from July 2018 to December 2019). Finally, the quality of English is rather low, and improvement is needed.
The section of Introduction is written relatively well, although I would recommend strengthening it by adding international sources. It would be great to highlight the international relevance of the study.
The design of the study is certainly unclear at present. This is not appropriate for readers to understand the matter of the study. Authors write "Participants were asked individually about their knowledge of the use of plant species for disease treatment.". However, we don't see what questions were inserted into the questionnaires. Therefore, there is no opportunity to estimate what exactly results were obtained during this study.
As said above, the section of Results and Discussion is actually the section of Results with certain exceptions. This is not appropriate for a so high-quality journal as Plants. That is why I recommend rejecting this manuscript. I recommend submitting this manuscript to certain African journals more oriented on the topic of this study.
The section of Conclusions is not connected with the obtained results and their discussion (to a certain degree due to the lack of discussion). Actually, this Conclusion could be used for any other paper on a similar topic. This is not suitable, I think.
Round 2
Reviewer 3 Report
Although the description of the methodology has been improved, the whole manuscript is still descriptive. No analysis is present in the section of Results. Table 1 is suitable rather for presenting in the Supplementing Materials, as it presents originally obtained data. They could be presented in analysis, but this was not done by the authors.
Unfortunately, the revision made by the authors was rather of cosmetic character. To my regret, these corrections have not solved the main problems.
Author Response
We humbly disagree that the results have not been analysed. Table 1 is a summary of the characteristics of the participants and relevant in the main manuscript and not as supplementary data.